# Spectral Library of Plant Species from Montesinho Natural Park in Portugal

Isabel Pôças [1,*], Cátia Rodrigues de Almeida [2,3,4], Salvador Arenas-Castro [5], João C. Campos [6], Nuno Garcia [6], João Alírio [2,3], Neftalí Sillero [6] and Ana C. Teodoro [2,3]

1. CoLAB ForestWISE—Collaborative Laboratory for Integrated Forest & Fire Management, Quinta de Prados, Campus da UTAD, 5001-801 Vila Real, Portugal
2. Department of Geosciences, Environment and Land Planning, Faculty of Sciences, University of Porto, Rua Campo Alegre, 687, 4169-007 Porto, Portugal; up201600831@fc.up.pt (C.R.d.A.); joao.m.alirio@gmail.com (J.A.); amteodor@fc.up.pt (A.C.T.)
3. Earth Sciences Institute (ICT), Pole of the FCUP, University of Porto, 4169-007 Porto, Portugal
4. Centro de Investigação de Montanha (CIMO), Instituto Politécnico de Bragança (IPB), Campus de Santa Apolónia, 5300-253 Bragança, Portugal
5. Area of Ecology—Department of Botany, Ecology and Plant Physiology, Faculty of Sciences, University of Cordoba, Campus de Rabanales, 14014 Córdoba, Spain; b62arcas@uco.es
6. CICGE—Centro de Investigação em Ciências GeoEspaciais, Faculdade de Ciências da Universidade do Porto, Alameda do Monte da Virgem, 4430-146 Vila Nova de Gaia, Portugal; jc.campos@fc.up.pt (J.C.C.); nuno.garcia@fc.up.pt (N.G.); neftali.sillero@gmail.com (N.S.)
* Correspondence: isabel.pocas@forestwise.pt

**Abstract:** In this work, we present and describe a spectral library (SL) with 15 vascular plant species from Montesinho Natural Park (MNP), a protected area in Northeast Portugal. We selected species from the vascular plants that are characteristic of the habitats in the MNP, based on their prevalence, and also included one invasive species: *Alnus glutinosa* (L.) Gaertn, *Castanea sativa* Mill., *Cistus ladanifer* L., *Crataegus monogyna* Jacq., *Frangula alnus* Mill., *Fraxinus angustifolia* Vahl, *Quercus pyrenaica* Willd., *Quercus rotundifolia* Lam., *Trifolium repens* L., *Arbutus unedo* L., *Dactylis glomerata* L., *Genista falcata* Brot., *Cytisus multiflorus* (L'Hér.) Sweet, *Erica arborea* L., and *Acacia dealbata* Link. We collected spectra (300–2500 nm) from five records per leaf and leaf side, which resulted in 538 spectra compiled in the SL. Additionally, we computed five vegetation indices from spectral data and analysed them to highlight specific characteristics and differences among the sampled species. We detail the data repository information and its organisation for a better understanding of the data and to facilitate its use. The SL structure can add valuable information about the selected plant species in MNP, contributing to conservation purposes. This plant species SL is publicly available in Zenodo platform.

**Keywords:** leaves spectra; spectral signatures; spectroradiometer; vascular plants

## 1. Introduction

Spectral libraries (SLs) encompassing visible (VIS), near-infrared (NIR), and shortwave infrared (SWIR) reflectance, up to 2.5 μm, are increasingly recognised as powerful and efficient tools to analyse and store large amounts of data on the properties of several Earth elements, such as vegetation, soil, and minerals (e.g., [1–11]).

Some examples of SLs currently available include the ECOSTRESS library [8], which integrates spectral data of vegetation and non-photosynthetic vegetation collected in the wavelength ranges of VIS-SWIR (0.35–2.5 μm) and thermal infrared (TIR, 2.5–15.4 μm); various national soil SLs, which include soil spectra and corresponding soil physical, chemical, and biological properties (e.g., [7,12,13]); the ASTER SL including over 2300 spectra of a wide variety of materials covering the wavelength range 0.4–15.4 μm (https://speclib.jpl.nasa.gov/, accessed on 1 December 2023 [14]); or the EcoSIS (https://ecosis.org, accessed on 1 December 2023 [15]) SL, which currently integrates around 200 datasets.

In particular, SLs of plant species can be used to (i) improve the understanding of biochemical, biophysical, and morphological plant properties (e.g., [10,16–18]); (ii) select the most sensitive and robust spectral features associated with unique species traits and support the classification of plant species and plant functional groups (e.g., [3,19]); and (iii) characterise the spectral variability over space and time while capturing changing phenological states, plant health condition, and environmental conditions (e.g., [4]), among others. Thus, SLs and their associated applications can provide information about the conditions, dynamics, and trends of plant species, which can be useful for planning management actions both for production and conservation purposes ([3,4,18]).

The spectral signatures of plants can be related to various plant properties because electromagnetic energy interacts with pigments, intercellular air spaces, and water within the plant's leaves, generating a specific pattern of reflectance throughout the wavelengths of the electromagnetic spectrum [20]. Overall, the spectral signatures of plants present a similar pattern, but each species has its specific spectral features, which can be described through SLs. Thus, rich documentation of the spectral data, including information on general dataset properties, data production information, repository information, and data reuse information, is paramount to leveraging the potential of data use for a broad range of applications, as discussed by [21].

We built a SL with a set of the most characteristic plant species in a conservation area in Portugal—Montesinho Natural Park (MNP)—under the framework of MontObEO project—Montesinho Biodiversity Observatory: an Earth Observation tool for biodiversity conservation (https://montobeo.wordpress.com/, accessed on 1 December 2023 [22]), funded by the Portuguese Foundation for Science and Technology (FCT: MTS/BRB/0091/2020 [23]). Specifically, we aimed to (i) characterise the general dataset properties and data production, (ii) describe the main patterns of target plants' spectral signatures, and (iii) present the repository information and SL organisation and its potential use.

This plant species SL is publicly available through the Zenodo platform, a research data repository, on the link https://doi.org/10.5281/zenodo.10798148, accessed on 1 December 2023 [24]. The Zenodo (https://about.zenodo.org/, accessed on 1 December 2023 [25]) is an open research data repository operated by CERN's Data Centre under the European OpenAIRE program, which assures safe data storage as long as CERN exists, and assigns a Digital Object Identifier (DOI) to make data citable and trackable. The SL will also be made available on the MontObEO project website.

## 2. Materials and Methods

### 2.1. Data Collection Area

We collected the data in MNP, located in the Northeast of Portugal (Figure 1). MNP is a protected area, classified also as a European Union's Natura 2000 site (https://natura2000.eea.europa.eu/Natura2000/SDF.aspx?site=PTCON0002, accessed on 1 December 2023 [26]). It lies between 6°30′53″ W and 7°12′9″ W longitude and between 41°43′47″ N and 41°59′24″ N latitude, in a territory with altitudes varying between 438 m and 1481 m, and high geological and climatic variability. It has a high diversity of vegetation and fauna [27], resulting from the diversity of habitats occurring in this mountainous area.

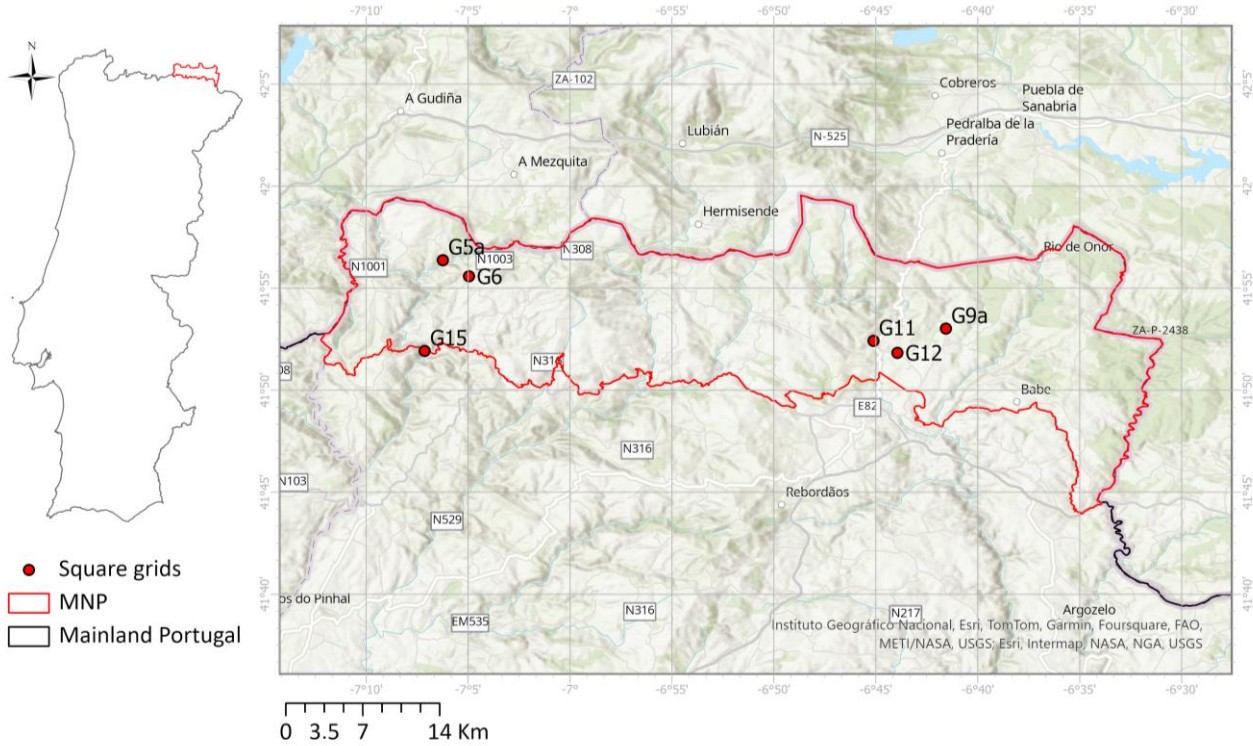

**Figure 1.** Data collection area at Montesinho Natural Park (MNP) with the distribution of the data plots (1 km × 1 km square grids). The centre of the square grids is illustrated for display purposes.

*2.2. Methodology*

The protocol for collecting the spectral data was based on the guidelines defined in [5,28]) and is described in the following subsections.

2.2.1. Sampling of Plant Species

The sampling focused on some of the most characteristic or relevant vascular plant species in the MNP based on previous work [27] and information from the Portuguese Institute for Nature Conservation and Forests (ICNF) (https://www.icnf.pt/, accessed on 1 December 2023 [29]). We identified more than 600 vascular plant species in the MNP (https://montobeo.shinyapps.io/MN-SPA_WebGIS/, accessed on 1 December 2023 [30]). We selected the most characteristic species considering the vascular plants typical of the habitats of the MNP, namely scrubland, natural meadows, chestnut groves, holm oak forests, riverside ecosystems, and oak woodlands (ICNF; https://www.icnf.pt/conservacao/rnapareasprotegidas/parquesnaturais/pnmontesinho, accessed on 1 December 2023 [29]).

For each species, we selected a minimum number of 10 presence points. We also included an invasive species (*Acacia dealbata* Link.) due to its high expansion risk inside the natural park. The final list includes 15 vascular plant species (Table 1). A brief characterisation of the selected species is presented in (Table 1).

We searched the selected plant species in six 1 km × 1 km square grids designated G5a, G6, G9a, G11, G12, and G15 (Figure 1; coordinates of the centre of the square grids in Table S1 of Supplementary Materials). We selected the six grid squares based on the species presence criterion, choosing the squares with the highest number of species. The six square grids were distributed over different areas of the MNP, to sample variable ecological and topogeographical conditions (e.g., related to orography and altitude). We considered prior information on the presence of the selected species in each sampling location to guide the collection of plant material.

**Table 1.** Vascular plant species selected for spectral data collection at Montesinho Natural Park (MNP), with a brief characterisation.

| Species | Family | Brief Characterisation * | European Conservation Status [31] | Species Code |
|---------|--------|--------------------------|-----------------------------------|--------------|
| *Acacia dealbata* Link. | Fabaceae | Trees up to 15 m, leaves evergreen, greyish-green, and flowers bright yellow. It occurs on the edges of forests, pine forests and thickets, as well as in the cool terrain of valleys, mountainous areas, banks of water courses, dunes, roadsides, and embankments. It invades, above all, after fires. | Not Evaluated (NE) | AcaDea |
| *Alnus glutinosa* (L.) Gaertn | Betulaceae | A riparian broadleaved tree species, common throughout Europe, with great plasticity in terms of climatic conditions whenever its roots are in almost permanent contact with a shallow water table. Normally grows up to 10–25 m tall, and presents dark green leaves measuring 4–10 cm. | Least Concern (LC) | AlnGlu |
| *Arbutus unedo* L. | Ericaceae | Shrub or small tree that can grow 8–10 m in height, but usually does not exceed 3–5 m. Presents simple leaves, slightly leathery, glabrous, glossy and dark green on the upper side, and paler on the underside. Very common in the Mediterranean basin, it spontaneously emerges in the understorey of cork oak, holm oak, and maritime pine stands in Portugal. | Least Concern (LC) | ArbUne |
| *Castanea sativa* Mill. | Fagaceae | This tree species is part of the deciduous oak forest and occurs throughout the central and western Mediterranean area. It can reach 20–30 m in height and the leaves are large, somewhat leathery, and deciduous to marcescent. | Least Concern (LC) | CastSat |
| *Cistus ladanifer* L. | Cistaceae | Evergreen shrub, which can exceed 2 m in height, with coriaceous, dark green leaves, and a large flower with 5 white petals. Occurs in xerophilic woodlands and thickets, under degraded cork or holm oak forests. | Not Evaluated (NE) | CistLan |
| *Crataegus monogyna* Jacq. | Rosaceae | Deciduous shrub or small tree, 2–5 m high, very branched and thorny, with simple, glabrous, dark green, glossy leaves on the upper side and matt green on the lower side. Inflorescence in corymbs; flowers with 5 free obovate petals, pinkish-white. | Not Evaluated (NE) | CratMon |
| *Cytisus multiflorus* (L'Hér.) Sweet | Fabaceae | Erect shrub, which can exceed 2 m in height, with angular and flexible branches, leaves fully developed after flowering and inflorescences with white flowers. Occurs from 120 to 1500 m of altitude, in siliceous substrates, on poor soils. | Least Concern (LC) | CytMul |
| *Dactylis glomerata* L. | Poaceae | Herbaceous plant, 10–100 (150) cm tall, most often with erect culms. Leaves are 2–8 (12) mm wide, dull green, flat, and 2–12 (20) mm long, and have inflorescences organised in panicles. It occurs in cool or shady places, grasslands in forest clearings, and in thickets and fallow areas in cork oak forests. | Not Evaluated (NE) | DactGlo |
| *Erica arborea* L. | Ericaceae | Tall, evergreen shrub, often exceeding 2 m in height, usually very branched from the base, with simple, glabrous, or sometimes pubescent leaves; solitary flowers or arranged in small umbels grouped in large numbers at the end of the branches. It occurs in scrubland and forest edges, in temperate or Mediterranean climates, near water lines, from sea level to 2000 m. | Least Concern (LC) | EriArb |

**Table 1.** *Cont.*

| Species | Family | Brief Characterisation * | European Conservation Status [31] | Species Code |
|---|---|---|---|---|
| *Frangula alnus* Mill. | Rhamnaceae | Small tree or shrub species, 3–5 m high, with erect trunk, sparse branches, simple, entire, alternate and petiolate leaves, with a shiny dark green upper leaf and a pale green lower leaf, glabrous or with pubescence on the nerves. It occurs in riparian woodland or riparian scrubland, hedgerows or under oak cover, on the banks of watercourses and ravines, always in damp places. | Least Concern (LC) | FranAln |
| *Fraxinus angustifolia* Vahl | Oleaceae | Deciduous tree that can reach 25 m in height, oval or rounded crown with numerous branches, and compound, opposite, subsessile leaves. It grows naturally in riparian woodland or deciduous woodland on mountain slopes. | Least Concern (LC) | FraxAng |
| *Genista falcata* Brot. | Fabaceae | Shrub 0.5–2 m tall, erect, with simple or compound thorns. It flowers and bears fruit from March to April. It appears on the edge of forests (oaks, chestnut groves) and open forest stands. Also in rocky places, on shale or granite, rarely on limestone. | Least Concern (LC) | GenFal |
| *Quercus pyrenaica* Willd. | Fagaceae | Tree up to 30 m tall, with an irregular crown. Very branched, with young branches and densely veltely-tomentose leaves. Deciduous to marcescent leaves: in drier seasons, typically Mediterranean or thermal, it presents marcescent behaviour, being deciduous in rainier areas or with greater proximity to the water table. | Least Concern (LC) | QuerPyr |
| *Quercus rotundifolia* Lam. | Fagaceae | Evergreen tree, up to 20 m in height, with a wide and irregular crown, adult trunk, and thick, non-suberous branches. Persistent leaves, dark green on the upper side, glabrescent with stellate hairs, greyish-green (glaucous) on the lower side, covered with stellate and fused-stellate hairs. It grows in skeletal, stony and rocky soils, poor in humus, with medium or low soil humidity, and it prefers regions with very hot and dry summers. | Least Concern (LC) | QuerRot |
| *Trifolium repens* L. | Fagaceae | Herbaceous perennial species, often occurring in wet meadows, mountain seminatural meadows, and on the banks of watercourses. | Least Concern (LC) | TrifRep |

* The main characteristics of the species are based on information from Virtual Biodiversity Museum, University of Évora (https://www.museubiodiversidade.uevora.pt/, accessed on 1 December 2023 [32]), and Botanical Garden from University of Trás-os-Montes e Alto Douro (Jardim Botânico UTAD, https://jb.utad.pt/, accessed on 1 December 2023 [33]).

2.2.2. Spectral Data Collection

We collected the spectral data on 21 and 22 June 2023, when the leaves of all target species were well-developed and at their peak of reflective radiation. We sampled five healthy leaves (*n* = 5) from adult plants per species in each square. We collected spectral data on both sides of individual leaves, except for *Acacia dealbata* Link., *Cytisus multiflorus* (L'Hér.) Sweet, *Dactylis glomerata* L., *Erica arborea* L., and *Genista falcata* Brot., where we considered a single side of the leaf for data collection due to the leaves' small size.

The data collection included the extraction of the leaves, a photographic record of at least one leaf from each species (shown in Figure S1 of Supplementary Materials), and the immediate acquisition of spectra using a spectroradiometer ASD FieldSpec® 4 (Figure 2). The spectroradiometer records spectral reflectance (after radiometric calibration), which represents how a specific object or surface reflects light at different wavelengths of the electromagnetic spectrum [20]. This spectroradiometer operates in the wavelength range of 300 to 2500 nm and rapidly collects data (approximately 0.2 s per spectrum). Its optical fibre is 1.5 m long with a 25° field of view, resulting in a spot size of 10 mm when using a contact

probe with an internal light source. The spectral resolution is 3 nm @ 700 nm and 10 nm @ 1400/2100 nm. Following the methodology adopted in other studies [1,34], we cleaned the lens of the contact probe with alcohol and turned on the equipment 30 min before its use, due to the detectors' sensitivity to temperature. The spectroradiometer was placed inside a vehicle, minimising both the effects of external light and vegetation degradation related to the delay between sample collection and spectral recording, which occurred within a few minutes.

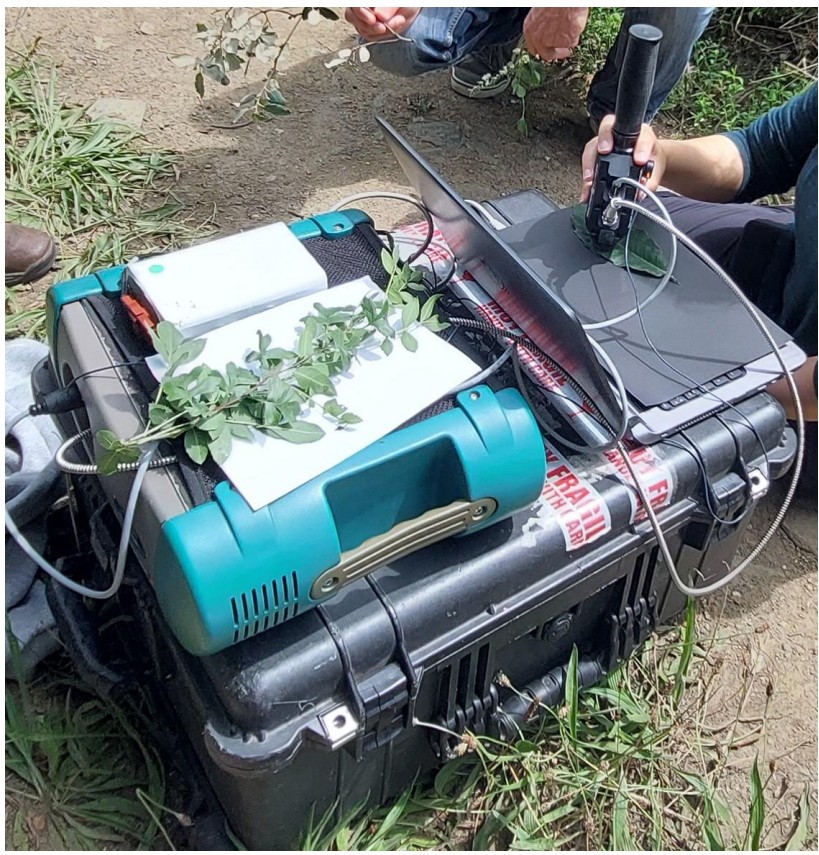

**Figure 2.** Reflectance data collection over a leaf, during fieldwork, with a spectroradiometer ASD FieldSpec® 4.

We obtained and recorded the spectra with the ASD ViewSpec Pro™ software, version 6.2. (ASD, Inc., Boulder, CO, USA) [35], performing the initial calibration of the equipment with a white reflectance standard. The calibration was repeated every time a new grid was sampled, when the equipment needed to be restarted, and as indicated by the software. The software was configured to acquire the spectrum of the leaves sampled from the average of ten measurements, aiming to enhance the signal-to-noise ratio.

We collected spectra from five records per leaf and leaf side (ventral and dorsal side of the leaf), totalling 2725 spectra (raw data), and averaged to obtain a spectrum per leaf and leaf side in each species, resulting in 538 spectra compiled in the SL. From the total of 2725 spectra collected, we excluded 35 due to noise, primarily associated with leaf size. All exclusions were related to the data collected on 22 June 2023, specifically involving *Cytisus multiflorus* (L'Hér.) Sweet (leaves 2, 3, 4, and 5 from grid G6, and leaf 4 from grid G5a) and *Genista falcata* Brot. (leaves 1 and 3 from grid G6).

### 2.2.3. Analysis of Spectral Data

To analyse the results visually and apply the necessary corrections to the data, we processed each spectral curve using the SpectraGryph software v1.2.16.1 [36]. For the construction of the SL, we used the 538 spectra, corresponding to one average spectra per

species, leaf, and leaf side in each grid. Subsequently, we employed a Python code from [37] to generate a .jpeg file containing the resulting spectral curves and further compiled the data into a SL stored in the Microsoft Access software, version 2403 Build 16. 0. 17425. 20176.

In addition to the collected data, we added five widely known vegetation indices (VIs) from literature to the SL, to assess specific characteristics of the samples, as follows:

(a)  Normalised Difference Vegetation Index (NDVI): sensitive to biomass, quantity, and condition of vegetation. NDVI values range from $-1$ to 1. Surfaces containing clouds or water result in an NDVI less than 0, while values close to 1 indicate healthier vegetation. This range also encompasses other landscape components (such as roads and construction) and highlights vegetation in the process of senescence [38]. The NDVI is calculated according to the Equation (1):

$$NDVI = (R800 - R680)/(R800 + R680) \tag{1}$$

where R800 represents the reflectance at 800 nm and R680 represents the reflectance at 680 nm, in the NIR and red domains, respectively.

(b)  Simple Ratio Vegetation Index (SR): highly sensitive to the presence of vegetation. It relies on the principle that leaves absorb relatively more energy in the red band than in the infrared band and thus SR will increase as the amount of leaves increases [39]. The SR is calculated according to the Equation (2):

$$SR = R800/R680 \tag{2}$$

where R800 represents the reflectance at 800 nm and R680 represents the reflectance at 680 nm.

(c)  Renormalised Difference Vegetation Index (RDVI): derived from NDVI, it is adapted to correct canopy saturation and linearise the relationship of vegetation biophysical properties by combining NIR and red bands [40]. The RDVI is calculated according to the Equation (3):

$$RDVI = (R800 - R670)/\sqrt{(R800 + R670)} \tag{3}$$

where R800 represents the reflectance at 800 nm and R670 represents the reflectance at 670 nm.

(d)  Greenness Index (GI): derived from a simple ratio between green and red bands, aiming to assess the overall health of vegetation [41,42]. The GI is calculated according to the Equation (4):

$$GI = R554/R667 \tag{4}$$

where R554 represents the reflectance at 554 nm and R667 represents the reflectance at 667 nm.

(e)  Structure Insensitive Pigment Index (SIPI): maximises the ratio between carotenoid and chlorophyll pigments, indicating an elevation in vegetation stress in high values [43]. The SIPI is calculated according to the Equation (5):

$$SIPI = (R800 - R445)/(R800 + R680) \tag{5}$$

where R800 represents the reflectance at 800 nm, R445 represents the reflectance at 445 nm and R680 represents the reflectance at 680 nm.

## 3. Results

### 3.1. Analysis of the Spectral Data

We sampled a total of 313 leaves for spectral data collection; 225 were sampled both on the ventral/front and dorsal/back sides (resulting in 450 samples) and 88 were sampled on a single side due to the small size of the leaves (Table 2).

**Table 2.** Number of square grids and leaves sampled per species. For *Acacia dealbata*, *Cytisus multiflorus*, *Dactylis glomerata*, *Erica arborea*, and *Genista falcata*, a single side of the leaf was considered for data collection due to its small size. For the other species, both the ventral/front and dorsal/back sides of the leaves were considered for data collection.

| Species | Number of Leaves Sampled | Number of Grids Sampled | Grids Designation |
|---|---|---|---|
| *Acacia dealbata* Link. | 5 | 1 | G6 |
| *Alnus glutinosa* (L.) Gaertn | 25 | 5 | G5a; G6; G11; G12; G15 |
| *Arbutus unedo* L. | 10 | 2 | G5a; G15 |
| *Castanea sativa* Mill. | 30 | 6 | G5a; G6; G9a; G11; G12; G15 |
| *Cistus ladanifer* L. | 25 | 5 | G6; G9a; G11; G12; G15 |
| *Crataegus monogyna* Jacq. | 25 | 5 | G5a; G9a; G11; G12; G15 |
| *Cytisus multiflorus* (L''Hér.) Sweet | 20 | 5 | G5a; G6; G9a; G12; G15 |
| *Dactylis glomerata* L. | 30 | 6 | G5a; G6; G9a; G11; G12; G15 |
| *Erica arborea* L. | 15 | 3 | G6; G12; G15 |
| *Frangula alnus* Mill. | 5 | 1 | G12 |
| *Fraxinus angustifolia* Vahl | 30 | 6 | G5a; G6; G9a; G11; G12; G15 |
| *Genista falcata* Brot. | 18 | 4 | G6; G9a; G11; G15 |
| *Quercus pyrenaica* Willd. | 25 | 5 | G5a; G6; G9a; G11; G12 |
| *Quercus rotundifolia* Lam. | 25 | 5 | G5a; G9a; G11; G12; G15 |
| *Trifolium repens* L. | 25 | 5 | G5a; G9a; G11; G12; G15 |

Figure 3 presents the average spectral curves recorded for the leaves of each one of 15 plant species sampled. The individual spectral curves recorded for each leaf (and side of the leaf) are presented in Figure S2 of Supplementary Materials and the detailed spectra are included in the SL available at https://doi.org/10.5281/zenodo.10798148, accessed on 1 December 2023 [24].

As observed in Figure 3, overall, the leaves' spectral reflectance curve is characterised by low values in the visible wavelengths, a drastic increase in values in the red edge region, high values in the NIR, and a decrease in the shortwave region, which is more pronounced around the 970 nm, 1190 nm, 1450 nm, and 1940 nm wavelengths.

Overall, the species *Cistus ladanifer* L., *Acacia dealbata* Link., *Erica arborea* L., *Cytisus multiflorus* (L'Hér.) Sweet, and *Genista falcata* Brot. presented lower reflectance values throughout the spectral range covered by the spectroradiometer (350–2500 nm), while the species *Quercus rotundifolia* Lam., *Crataegus monogyna* Jacq., and *Fraxinus angustifolia* Vahl presented higher reflectance values (Figure 3).

Regarding the species with spectral data recorded on both sides of the leaves, the average difference of reflectance values between the ventral/front side and the dorsal/back side is presented in Table 3, per region of the electromagnetic spectrum. The species *Alnus glutinosa* (L.) Gaertn, *Castanea sativa*, *Crataegus monogyna* Jacq., *Fraxinus angustifolia* Vahl, *Quercus pyrenaica* Willd., *Quercus rotundifolia* Lam., and *Arbutus unedo* L. presented the largest average differences of reflectance between both sides of the leaves in the visible range of the electromagnetic spectrum, while the species *Cistus ladanifer* L. and *Frangula alnus* Mill. had the largest average differences in the SWIR region (Table 3).

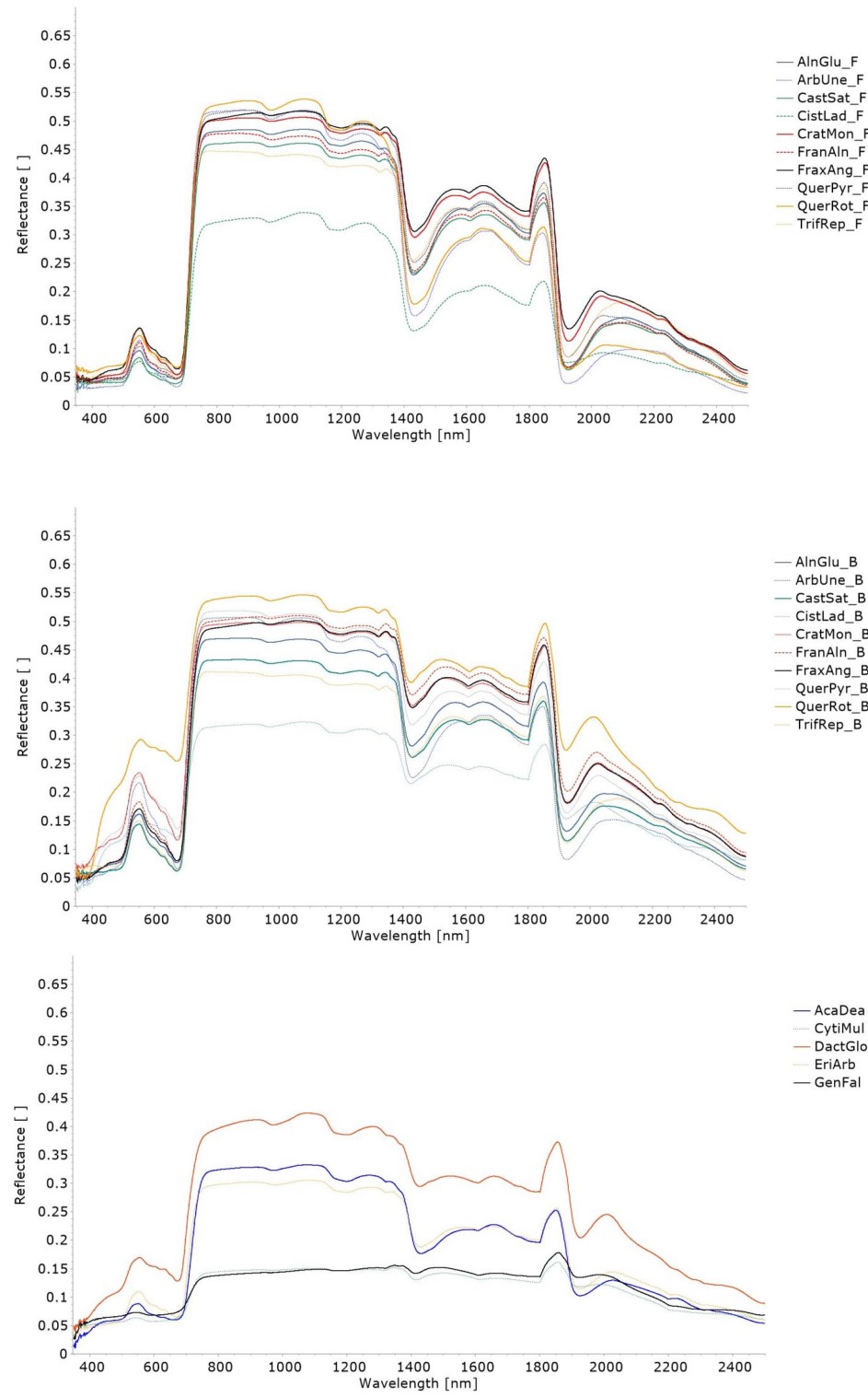

**Figure 3.** Average spectral curves, covering the range 350–2500 nm, collected over the leaves of the 15 plant species sampled: (**top**) ventral/front (F) side; (**middle**) dorsal/back (B) side; (**bottom**) single side. Both sides of the leaves were considered for *Alnus glutinosa* (L.) Gaertn (AlnGlu), *Castanea sativa* Mill. (CatSat), *Cistus ladanifer* L. (CistLad), *Crataegus monogyna* Jacq. (CratMon), *Frangula alnus* Mill. (FranAln), *Fraxinus angustifolia* Vahl (FraxAng), *Quercus pyrenaica* Willd. (QuerPyr), *Quercus rotundifolia* Lam. (QuerRot), *Trifolium repens* L. (TrifRep), and *Arbutus unedo* L. (ArbUne), while a single side of the leaves was considered for *Dactylis glomerata* L. (DactGlo), *Genista falcata* Brot. (GenFal), *Cytisus multiflorus* (L'Hér.) Sweet (CytMult), *Acacia dealbata* Link. (AcaDea), and *Erica arborea* L. (EriArb).

**Table 3.** Average reflectance difference between both sides of the leaves per region of the electromagnetic spectrum and species.

|  | Visible (300–670 nm) | Red Edge (670–780 nm) | NIR (780–1000 nm) | SWIR (1000–2500 nm) |
|---|---|---|---|---|
| *Alnus glutinosa* (L.) Gaertn | −0.034 | −0.012 | 0.031 | −0.010 |
| *Castanea sativa* Mill. | −0.050 | −0.013 | 0.050 | −0.002 |
| *Cistus ladanifer* L. | −0.063 | −0.063 | −0.036 | −0.070 |
| *Crataegus monogyna* Jacq. | −0.057 | −0.033 | 0.007 | −0.024 |
| *Frangula alnus* Mill. | −0.038 | −0.020 | 0.003 | −0.053 |
| *Fraxinus angustifolia* Vahl | −0.034 | −0.024 | 0.010 | −0.012 |
| *Quercus pyrenaica* Willd. | −0.109 | −0.091 | −0.019 | −0.059 |
| *Quercus rotundifolia* Lam. | −0.078 | −0.041 | 0.050 | −0.075 |
| *Trifolium repens* L. | −0.025 | −0.020 | −0.004 | 0.004 |
| *Arbutus unedo* L. | −0.055 | −0.032 | 0.011 | −0.032 |

Regarding the individual spectral curves recorded for each leaf per species, *Cistus ladanifer* L., *Dactylis glomerata* L., *Fraxinus angustifolia* Vahl, and *Erica arborea* L. showed the largest variability among spectra, while *Castanea sativa*, *Frangula alnus* Mill., *Arbutus unedo* L., *Quercus pyrenaica* Willd., and *Alnus glutinosa* (L.) Gaertn presented the lowest variability (Figure S1 of Supplementary Materials).

The SL publicly available on https://doi.org/10.5281/zenodo.10798148, accessed on 1 December 2023 [24] also includes a set of vegetation indices (Equations (1)–(5)) computed for each species (Figure 4).

Overall, the highest vegetation index values corresponded to *Arbutus unedo* L. (ArbUne), *Alnus glutinosa* (L.) Gaertn (AlnGlu), and *Castanea sativa* Mill. (CatSat), while the lowest values corresponded to the species *Genista falcata* Brot. (GenFal), and *Cytisus multiflorus* (L'Hér.) Sweet (CytMult).

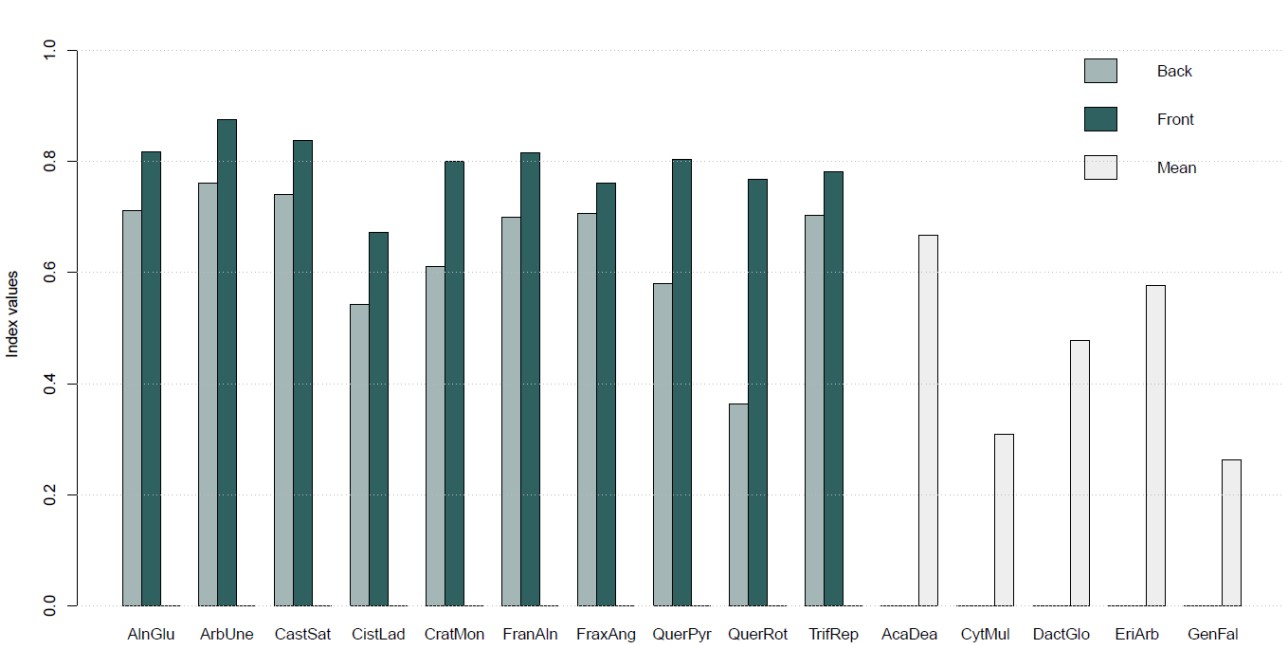

**Figure 4.** *Cont.*

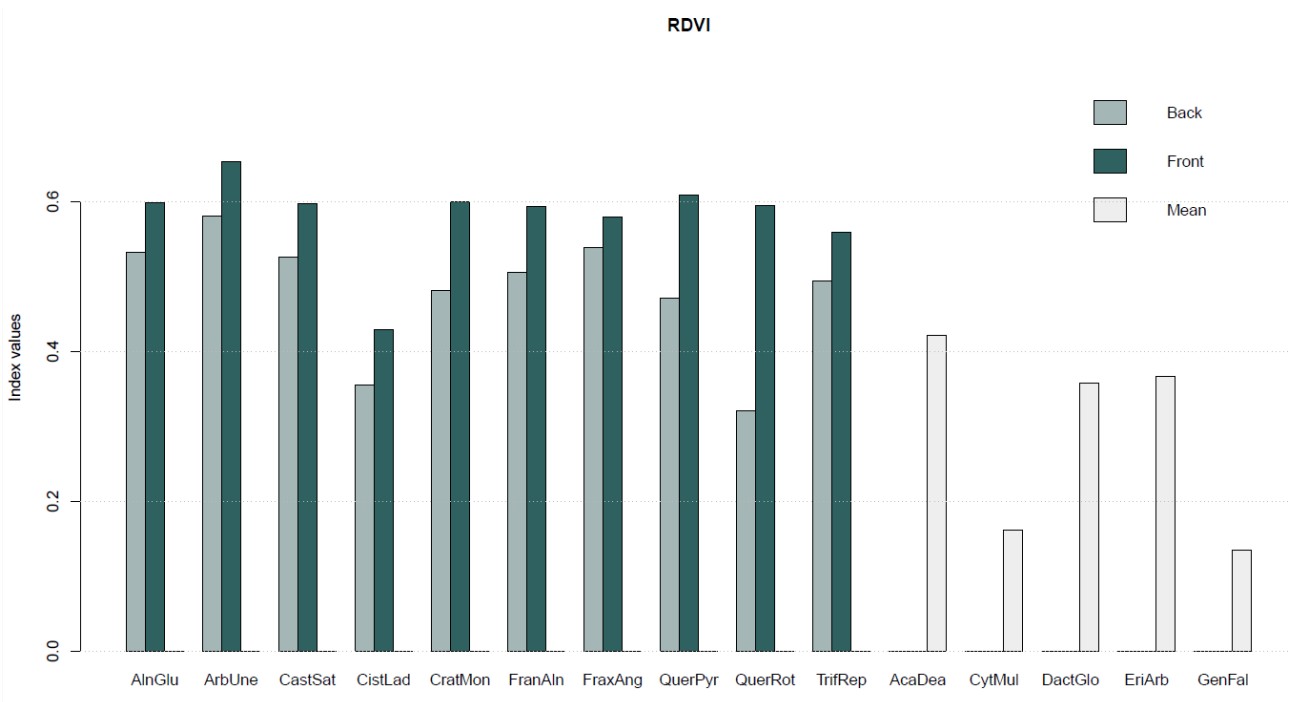

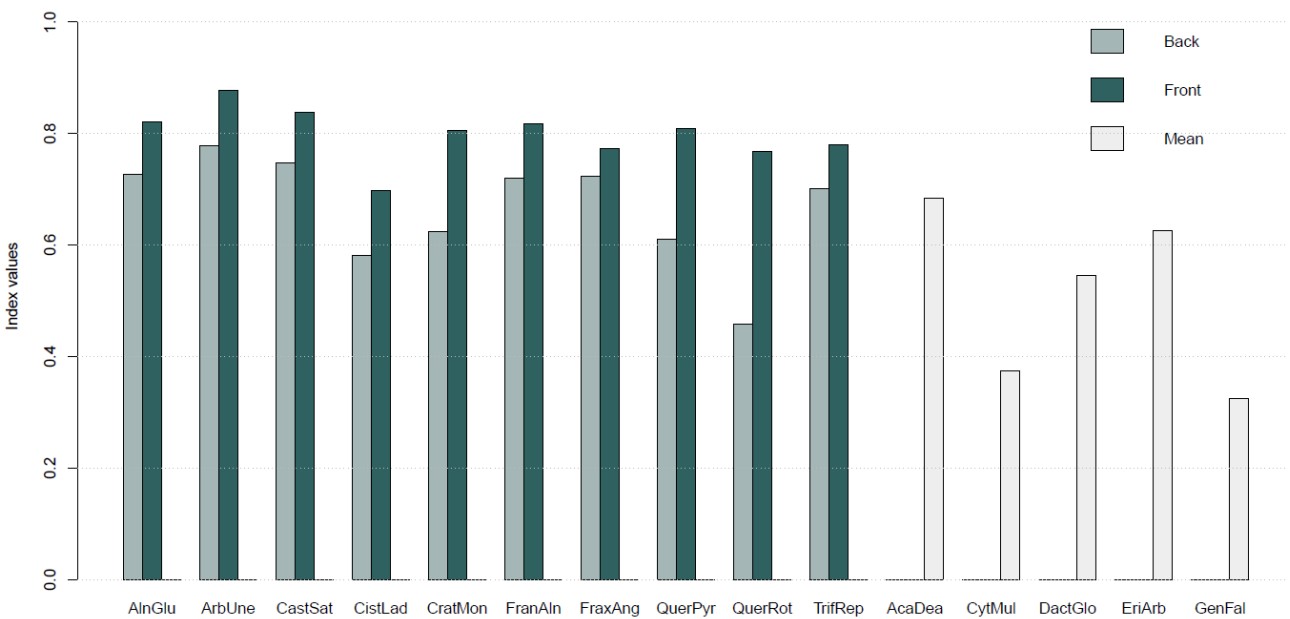

**Figure 4.** *Cont.*

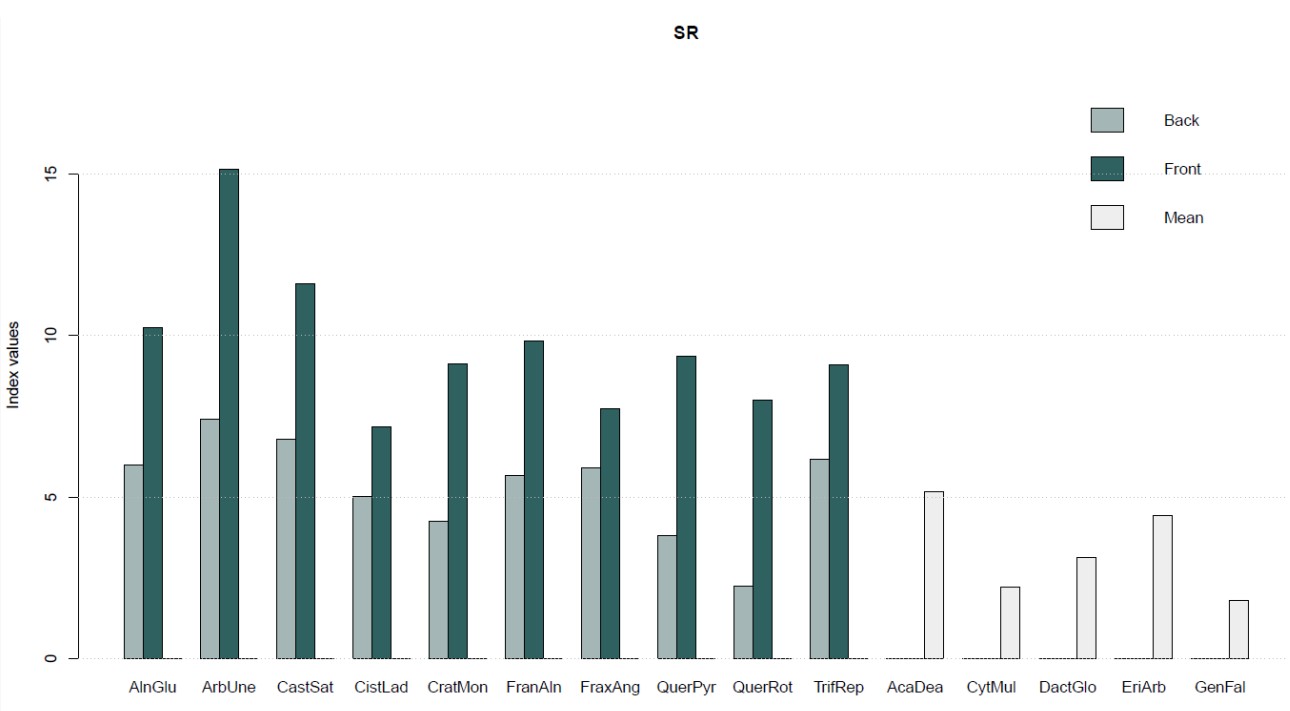

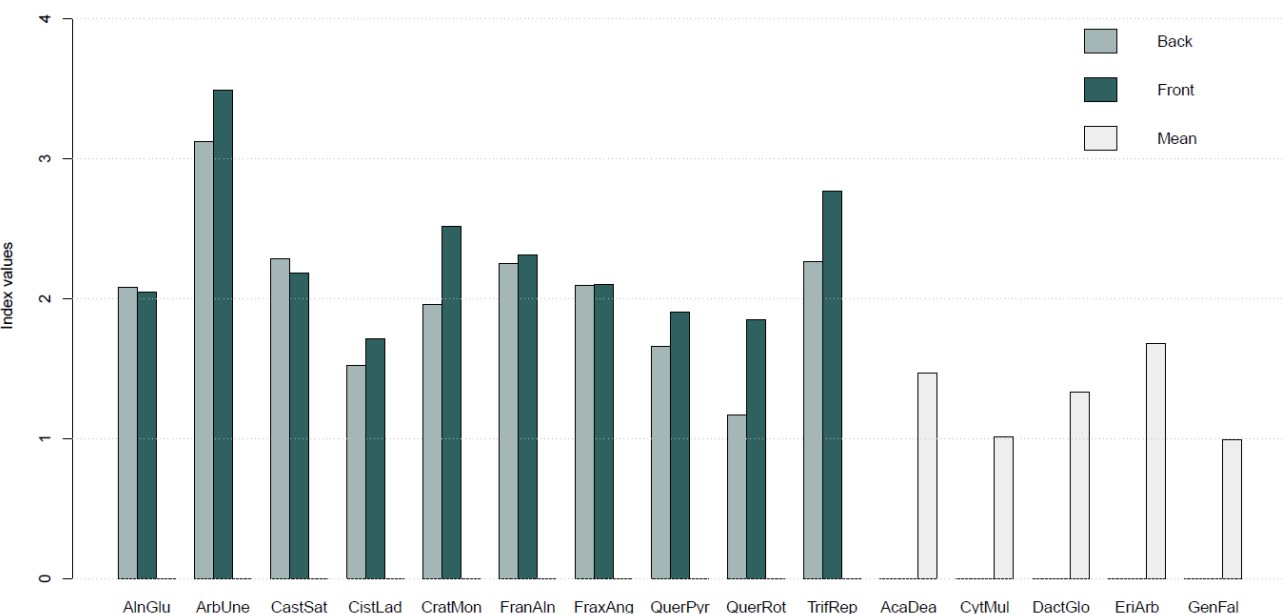

**Figure 4.** Average values of vegetation indices—Normalised Difference Vegetation Index (NDVI), Renormalised Difference Vegetation Index (RDVI), Structure Insensitive Pigment Index (SIPI), Simple Ratio Vegetation Index (SR), and Greenness Index (GI)—computed for each plant species. For the species where both sides of the leaf were sampled, the vegetation indices values are presented per side of the leaf (back and front), while in the remaining species only a mean value is presented.

### 3.2. Data Repository Information and Organisation

The general information and organisation of the SL in the repository are described in Table 4, detailing all the fields included and their description.

**Table 4.** Description of the information contained in the spectral library (SL).

| Field | Description | Attachments Files |
|---|---|---|
| ID | Primary key created by the database | - |
| Date | Date of sampling | - |
| Spectrum_name | Spectrum name, with species indication, leaf number, grid, collection number, and sampled leaf side, for example: AlnGlu_L1_G11_C1_B where 'AlnGlu' refers to the species *Alnus glutinosa* (L.) Gaertn, 'L1' refers to leaf 1, 'G11' is grid 11, 'C1' is collection 1, and 'B' refers to the back. | - |
| Grid | Location identification where the sample was collected | - |
| Specie | Identification of the sampled species | - |
| Side | Sampled side of the plant (Back, Front, or Single Side) | - |
| Latitude | Y-coordinate in UTM (Northing) | - |
| Longitude | X-coordinate in UTM (Easting) | - |
| Photo | Sample photograph | .jpg .pdf .csv |
| Spectra | Final average of spectra calculated from five raw files, obtained in the .asd extension. | .jpg .pdf .csv |
| Normalized_Difference_Vegetation_Index_NDVI | Calculated result of NDVI | - |
| Simple_Ratio_SR | Calculated result of SR | - |
| Renormalized_Difference_Vegetation_Index_RDVI | Calculated result of RDVI | - |
| Greenness_Index_GI | Calculated result of GI | - |
| Structure_Insensitive_Pigment_Index_SIPI | Calculated result of SIPI | - |

## 4. Discussion

The spectral reflectance curves recorded for the leaves sampled in each species (Figure 3) follow typical patterns for the vegetation, characterised by low values in the visible wavelengths and high values in the NIR ([20,44]). The low reflectance of the leaves in the visible domain is due to the strong absorption of incident energy by the leaf pigments in the palisade mesophyll [45]. The high reflectance in the NIR domain is due to the scattering from internal leaf structures, specifically the cells and intercellular air spaces in the spongy mesophyll layer, where the exchanges of oxygen and carbon dioxide associated with photosynthesis and respiration occur [20]. Around 700 nm wavelength, single leaves of most species present a sharp increase in reflectance curve [45]—red edge region—as also observed in this work. The decrease in reflectance in the shortwave spectral domain (Figure 3) is more pronounced around the 970 nm, 1190 nm, 1450 nm, and 1940 nm wavelengths, which correspond to liquid-water absorption bands.

The spectral patterns recorded are relative to individual leaves, as in other SLs containing vegetation spectra (e.g., ECOSTRESS SL [8]). The reflectance response of the canopy can deviate from the reflectance of individual leaves due to the canopy structure, with various associated elements (e.g., leaves, stems, fruits) and inherent arrangements, the optical properties of the soil underneath, and the interactions of the incident radiation with the vegetation canopy, as discussed by several authors (e.g., [8,46,47]). Nevertheless, radiative transfer models can be used to scale leaf level measurements to canopy level (e.g., [48,49]).

Differences in leaves' reflectance along the electromagnetic spectrum (Figure 3) can result from variations in biophysical, biochemical, and morphological features, including

leaf thickness, leaf structure, water content, nitrogen content, and fibre constituents, as discussed by various authors (e.g., [48,50,51]). In order to translate spectral reflectance into meaningful information about plant species, specific plant traits, or plant conditions (e.g., related to stress), several modelling and retrieval approaches can be used: methods relating spectral bands, vegetation indices (VIs), or spectral ratios with plant functional traits; physically based model inversion methods establishing a cause–effect relationship grounded on physical knowledge; and hybrid methods [52]. Spectral information along the electromagnetic spectrum, obtained from SLs, can feed such retrieval methods to assess the dynamics, condition, and health of vegetation, and then support management decisions. Additionally, spectral information obtained from SLs can provide useful insights for the classification of plant species and validate information derived from satellite imagery.

In addition, SLs are progressively being recognised as valuable tools to support current and future hyperspectral missions, such as the Hyperspectral Precursor of the Application Mission (PRISMA mission, https://www.asi.it/en/earth-science/prisma/, accessed on 1 December 2023 [53]), launched in March 2019, the Environmental Mapping and Analysis Program (EnMAP mission, https://www.enmap.org/, accessed on 1 December 2023 [54]) launched in April 2022, and the forthcoming Copernicus Hyperspectral Imaging Mission for the Environment (CHIME). Also, data from SLs are key for the development and assessment of new portable devices, useful for in situ observations, as discussed by [9].

## 5. Conclusions

Our data can improve data provenance information and enhance the potential for data reuse. We highlighted the reflectance response of each species along the range between 350 and 2500 nm of the electromagnetic spectrum. The SL of plant species characteristic for a conservation area in Portugal (MNP) can add valuable information, contributing to conservation purposes.

**Supplementary Materials:** The following supporting information can be downloaded at: https://www.mdpi.com/article/10.3390/data9050065/s1, Figure S1. Photographic record of a leaf from each vascular plant species selected for spectral data collection. Figure S2. Individual spectral curves, covering the range 350–2500 nm, collected per leaf and leaf side in each species. Both sides of the leaves (ventral/front (F) side and forsal/back (B) side) were considered for *Alnus glutinosa* (L.) Gaertn (AlnGlu), *Arbutus unedo* L. (ArbUne), *Castanea sativa* Mill. (CatSat), *Cistus ladanifer* L. (CistLad), *Crataegus monogyna* Jacq. (CratMon), *Frangula alnus* Mill. (FranAln), *Fraxinus angustifolia* Vahl (FraxAng), *Quercus pyrenaica* Willd. (QuerPyr), *Quercus rotundifolia* Lam. (QuerRot), and *Trifolium repens* L. (TrifRep), while a single side of the leaves was considered for *Acacia dealbata* Link. (AcaDea), *Cytisus multiflorus* (L'Hér.) Sweet (CytMult), *Dactylis glomerata* L. (DactGlo), *Erica arborea* L. (EriArb), and *Genista falcata* Brot. (GenFal). Table S1. Coordinates (WGS 84) of the centre of the square grids sampled at Montesinho National Park.

**Author Contributions:** Conceptualisation, I.P., S.A.-C., A.C.T. and N.S.; methodology, I.P., S.A.-C., A.C.T. and N.S.; investigation, I.P., S.A.-C., C.R.d.A., A.C.T., N.S., J.C.C., N.G. and J.A.; writing—original draft preparation, I.P. and C.R.d.A.; writing—review & editing, I.P., S.A.-C., C.R.d.A., A.C.T., N.S., J.C.C., N.G. and J.A.; project administration, N.S.; funding acquisition, N.S., S.A.-C., A.C.T. and I.P. All authors have read and agreed to the published version of the manuscript.

**Funding:** This research was supported by Portuguese national funds through FCT—Portuguese Foundation for Science and Technology I.P., under MontObEO—Montesinho biodiversity observatory: an Earth Observation tool for biodiversity conservation (FCT: MTS/BRB/0091/2020). Cátia Rodrigues de Almeida was financially supported by Portuguese national funds through FCT—Foundation for Science and Technology I.P. (Grant: PRT/BD/153518/2021). Neftalí Sillero is supported by a CEEC2017 contract (CEECIND/02213/2017) from FCT. João C. Campos and Nuno Garcia are supported respectively by a research contract and grants from MontObEO project (MTS/BRB/0091/2020). Salvador Arenas-Castro is supported by a María Zambrano fellowship funded by the Spanish Ministry of Universities and the European Union-Next Generation Plan.

**Institutional Review Board Statement:** Not applicable.

**Informed Consent Statement:** Not applicable.

**Data Availability Statement:** The plant species spectral library (SL) described in the present manuscript is publicly available on the link https://doi.org/10.5281/zenodo.10798148, accessed on 1 December 2023 [24].

**Conflicts of Interest:** The authors declare no conflicts of interest.

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
