# Peer review of "Spectral Library of Plant Species from Montesinho Natural Park in Portugal"

_data_

Round 1
Reviewer 1 Report
Comments and Suggestions for Authors
This is a well-written manuscript and adds important value addition to the spectral library of plant species native to the region of interest.
I only have the following minor suggestions and comments that will help improve the manuscript.
Figure 1. Please include lat-lon extent to the image boundaries for clarity. I understand you have written in the text – but having them in the image helps in clarity and rapid reproduction of the shapefiles.
I could not access the zenodo website as the application was not available (I hope its temporary). Did you also provide the point and area shapefiles of the regions marked in Fig.1 ? Please do so, if you haven’t already – as that will help in quick identification on ground truth labels for satellite remote sensing studies.
Line 138: You have a photographic record of each species. Please show them in a figure within the manuscript for their quicker identification in the field by future researchers
Line 157: what do you mean by leaf and leaf side? Do you mean ventral and dorsal side of the leaves? Please show these by example of a picture.
Figure 2. The colors used in the top and middle figure of Figure 2 have very similar colors in the same plot – that is very confusing - for eg (ArbUne_B and QuerPyr_B). This is very difficult for the readers especially with color blindness. Please make sure you differentiate these spectra by splitting the plots or by adding symbols within the line plot to aid differentiation
Section 2.2.3: How did you come up with these indices? Did you take it from previous literature? Or you formulated to facilitate the detection of these specific species? If you did formulate them, help the readers by adding the spectral band center markers in the SL plot so that readers can see the importance of these chosen bands.
Lines 262-268 and Table 3 shows that there is variability in reflectance values between their front and back sides for each species. In that case why did you take the average of the index values between front and back side for your bar chart in Figure 3? Plot the corresponding front and back side index values for each species in the bar chart – this will help us to learn how important and different/closer these index values for front and back side of each species. please also add the exact values of each index for each species at the end of its specific bar chart value for clarify.
Author Response
Reviewer 1
R#1: This is a well-written manuscript and adds important value addition to the spectral library of plant species native to the region of interest. I only have the following minor suggestions and comments that will help improve the manuscript.
R: We appreciate the Reviewer's comments. We respond below to each of the Reviewer's suggestions and comments. Changes to the manuscript are marked in the revised manuscript with track changes.
R#1: Figure 1. Please include lat-lon extent to the image boundaries for clarity. I understand you have written in the text – but having them in the image helps in clarity and rapid reproduction of the shapefiles.
R: We revised Figure 1 was revised, and to include included a grid with lat-lon information on the image boundaries was included.
R#1: I could not access the zenodo website as the application was not available (I hope its temporary). Did you also provide the point and area shapefiles of the regions marked in Fig.1 ? Please do so, if you haven’t already – as that will help in quick identification on ground truth labels for satellite remote sensing studies.
R: We have tested the link to the spectral library in Zenodo in different browsers and it is working. The Reviewer's difficulty in accessing the link was probably due to a temporary glitch. We provided in the Zenodo link/webpage the coordinates (WGS 84) of the centre of the square grids (1 km x 1 km) sampled at Montesinho National Park in the dataset’s report along with the spectral library.
R#1: Line 138: You have a photographic record of each species. Please show them in a figure within the manuscript for their quicker identification in the field by future researchers
R: The photographic record of a leaf from each species considered in this study is now provided in Figure S1 of Supplementary Material.
R#1: Line 157: what do you mean by leaf and leaf side? Do you mean ventral and dorsal side of the leaves? Please show these by example of a picture.
R: In the revised manuscript, we have clarified this issue as follows: “We collected spectra from five records per leaf and leaf side (ventral and dorsal side of the leaf), totalling 2725 spectra (raw data), and averaged to obtain a spectrum per leaf and leaf side in each species, resulting in 538 spectra compiled in the SL.“ Also, throughout the manuscript, we replaced “upper/front side” by “ventral/front side” and “lower/front side” by “dorsal/back side” for additional clarification.
R#1: Figure 2. The colors used in the top and middle figure of Figure 2 have very similar colors in the same plot – that is very confusing - for eg (ArbUne_B and QuerPyr_B). This is very difficult for the readers especially with color blindness. Please make sure you differentiate these spectra by splitting the plots or by adding symbols within the line plot to aid differentiation.
R: We provide the colour and patterns of the graph lines of Figure 3 (Figure 2 in the original version of the manuscript) for better differentiation of the spectra, following the Reviewer's suggestion.
R#1: Section 2.2.3: How did you come up with these indices? Did you take it from previous literature? Or you formulated to facilitate the detection of these specific species? If you did formulate them, help the readers by adding the spectral band center markers in the SL plot so that readers can see the importance of these chosen bands.
R: We selected the vegetation indices from the literature. The references to the corresponding literature, for each vegetation index, are indicated in the manuscript. We added additional information for better clarification: "we added five widely known Vegetation Indices (VIs) from literature to the SL, to assess specific characteristics of the samples, as follows"
R#1: Lines 262-268 and Table 3 shows that there is variability in reflectance values between their front and back sides for each species. In that case why did you take the average of the index values between front and back side for your bar chart in Figure 3? Plot the corresponding front and back side index values for each species in the bar chart – this will help us to learn how important and different/closer these index values for front and back side of each species. please also add the exact values of each index for each species at the end of its specific bar chart value for clarify.
R: We updated Figure 4 in the revised version of the manuscript (Figure 3 in the original version of the manuscript): now, it includes the VIs of the front and back sides for each species in the bar charts.
Reviewer 2 Report
Comments and Suggestions for Authors
This is an excellent paper. It is so nice to read a well-written and well-structured paper presented in a finished form! Well done. The paper is very interesting and presents an up-to-date approach to developing a reference of plant leaf reflectance which has many uses - as indicated in the paper. The paper is very well-documented and referenced with good detail etc. I see quite a few references to reflectance models in the reference list which is good, and mention of e.g. bidirectional reflectance etc. I wondered if the authors might add a little more in about e.g. spectral reflectance, albedo/spectral albedo, directional and bidirectional reflectance as well as plant canopy models also to the text - maybe just a short paragraph - as there was quite a lot of work done on spectral reflectance in a plant/agricultural and horticultural sciences to develop databases/banks and latterly with mathematical models of plant leaf and canopy reflectance where such data can play a role in validation/verification, as well as satellite image studies for classification of imagery etc. I also wondered if there is a bit more detail on what the instrument measures, as well as some images/diagrams of the instrument and its use in the field etc., together with some additional photographs for illustration of some of the plant characteristics. This sort of information is always useful to the reader. Overall a very interesting read and I am glad to see such work being carried out, and to be exposed to a very nicely documented piece of research.
Author Response
Reviewer 2
R#2: This is an excellent paper. It is so nice to read a well-written and well-structured paper presented in a finished form! Well done. The paper is very interesting and presents an up-to-date approach to developing a reference of plant leaf reflectance which has many uses - as indicated in the paper. The paper is very well-documented and referenced with good detail etc.
R: We appreciate the Reviewer's comments. We respond below to each of the Reviewer's suggestions and comments. Changes are marked in the revised manuscript with track changes.
R#2: I see quite a few references to reflectance models in the reference list which is good, and mention of e.g. bidirectional reflectance etc. I wondered if the authors might add a little more in about e.g. spectral reflectance, albedo/spectral albedo, directional and bidirectional reflectance as well as plant canopy models also to the text - maybe just a short paragraph - as there was quite a lot of work done on spectral reflectance in a plant/agricultural and horticultural sciences to develop databases/banks and latterly with mathematical models of plant leaf and canopy reflectance where such data can play a role in validation/verification, as well as satellite image studies for classification of imagery etc.
R: Following the Reviewer's suggestion, we added in section 2.2.2 additional information relative to surface reflectance: "The spectroradiometer records spectral reflectance (after radiometric calibration), which represents how a specific object or surface reflects light at different wavelengths of the electromagnetic spectrum [21]". Also, we added to the discussion section additional information relative to retrieval methods for obtaining biophysical variables based on spectral data, and the usefulness of reflectance data from SL: “In order to translate spectral reflectance into meaningful information about plant species, specific plant traits or plant conditions (e.g., related to stress), several modelling and retrieval approaches can be used: methods relating spectral bands, vegetation indices (VIs) or spectral ratios with plant functional traits; physically-based model inversion methods establishing a cause-effect relationship grounded on physical knowledge; and hybrid methods [53]. Spectral information along the electromagnetic spectrum, obtained from SLs, can feed such retrieval methods to assess the dynamics, condition, and health of vegetation, and then support management decisions. Additionally, spectral information obtained from SLs can provide useful insights for the classification of plant species and validate information derived from satellite imagery.”
R#2: I also wondered if there is a bit more detail on what the instrument measures, as well as some images/diagrams of the instrument and its use in the field etc., together with some additional photographs for illustration of some of the plant characteristics. This sort of information is always useful to the reader.
R: We added in section 2.2.2 additional information relative to what the spectroradiometer measures: " The spectroradiometer records spectral reflectance (after radiometric calibration), which represents how a specific object or surface reflects light at different wavelengths of the electromagnetic spectrum [21]". Additionally, we added a new Figure to the manuscript, showing the spectroradiometer measuring a leaf in the field (Figure 2).
The photographic record of a leaf from each species is now provided in Figure S1 of Supplementary Material.
R#2: Overall a very interesting read and I am glad to see such work being carried out, and to be exposed to a very nicely documented piece of research.
R: We appreciate the Reviewer's comments.